# Research on Risk Detection of Autonomous Vehicle Based on Rapidly-Exploring Random Tree

**Yincong Ma, Kit Guan Lim \*, Min Keng Tan** **, Helen Sin Ee Chuo, Ali Farzamnia** **and Kenneth Tze Kin Teo \***

Faculty of Engineering, Universiti Malaysia Sabah, Kota Kinabalu 88400, Malaysia
\* Correspondence: limkitguan@ums.edu.my (K.G.L.); kenteo@ums.edu.my (K.T.K.T.)

**Abstract:** There is no doubt that the autonomous vehicle is an important developing direction of the auto industry, and, thus, more and more scholars are paying attention to doing more research in this field. Since path planning plays a key role in the operation of autonomous vehicles, scholars attach great importance to this field. Although it has been applied in many fields, there are still some problems, such as low efficiency of path planning and collision risk during driving. In order to solve these problems, an automotive vehicle-based rapid exploration random tree (AV-RRT)-based non-particle path planning method for autonomous vehicles is proposed. On the premise of ensuring safety and meeting the requirements of the vehicle's kinematic constraints through the expansion of obstacles, the dynamic step size is used for random tree growth. A non-particle collision detection (NPCD) collision detection algorithm and path modification (PM) path modification strategy are proposed for the collision risk in the turning process, and geometric constraints are used to represent the possible security threats, so as to improve the efficiency and safety of vehicle global path driving and to provide reference for the research of driverless vehicles.

**Keywords:** auto drive; risk detection; path planning; rapidly-exploring random tree



## 1. Introduction

It is indisputable that autonomous vehicles are is an important developing direction in the auto industry, and they enable an automobile to arrive at its destination safely through a self-driving system. At present, an autonomous vehicle mainly refers to environment awareness and path planning. In particular, environment awareness is the basis for path planning which indicates that the implementation of path planning is based on the result of environment awareness.

Although environment awareness is applied in many fields, it still has some factors which must be considered, such as the high cost of data labeling and lower data precision. In order to solve these problems, foreign and domestic scholars have attached much importance to carrying out in-depth research. Some scholars have proposed the use of potential field path planning technology, which is convenient for real-time control at the bottom. It has been widely studied in real-time obstacle avoidance and smooth track control of autonomous vehicles. However, the artificial potential field path planning method usually has local minima [1]. Some scholars use model predictive control, which has the basic characteristics of a predictive model, rolling optimization, and feedback correction, and is suitable for control systems that are not easy to establish accurate mathematical models for and which have constraints. However, at present, the precision of path tracking of model predictive control at high speed is difficult to guarantee [2]. Typical graph search algorithms include the Dijkstra algorithm, A* algorithm, D* algorithm, etc. This kind of algorithm represents the environment space with a graph, and then converts the path planning into the traversal of graph vertices. This kind of algorithm searches out relatively good paths and has high efficiency and stability. The disadvantage is that in complex environment spaces, memory consumption is huge [3]. The most studied algorithm is the

rapidly-exploring random tree algorithm. This kind of algorithm realizes the coverage search of the environment space by randomly selecting and adding sampling points in the state space. Its advantages are that it does not need modeling, and the search speed is fast. It is suitable for high-dimensional space and complex environment spaces, but its disadvantages are that there are many redundant points, it consumes large storage space, and its stability is poor. To solve these problems, this paper is devoted to putting forwards a new approach for risk detection of autonomous vehicles based on the rapidly-exploring random tree (RRT) approach, aimed at reducing the cost of high-precision data labeling. This approach relies on judgement of the distance between each position of the target automobile in its path and sample. When the position and sample are in a same subspace, the image might be processed with rough sort, and when they are in different subspace, the image might be processed according to distinguishing the subspace where the position is located from where the sample is located. Adopting this approach is not only conducive to raising the precision of data, but also contributes to reducing the cost of data labeling. Throughout all the experiments conducted in this research, this paper proves the feasibility of this approach. In addition, an algorithm for path planning not only refers to the probabilistic roadmap method (PRM) [4] and RRT but is also based on entertainment awareness. This paper discusses the technology of improving the risk detection of an autonomous vehicle based on RRT, aiming at putting forward the risk detection idea of the AV-RRT algorithm, and proposing a vehicle-oriented non-particle collision detection method to reduce the risk of path turning, and then modifying the path of the dangerous paths in collision detection with certain strategies to avoid the collision risk that may occur in the actual driving process of vehicles in advance [5]. As a result, the security of the global path is improved.

The rest of the paper is organized as follows. The second section introduces the basic RRT algorithm and proposes an AV-RRT algorithm for global path planning of autonomous vehicles to solve these problems, aiming at the problems that do not conform to kinematic constraints, poor practicability, and low planning efficiency. According to the possible risks of the proposed AV-RRT algorithm, the PM path modification strategy and NPCD risk detection method are designed and proposed in Sections 3 and 4, respectively, to improve the security of the global path. The fifth section is the simulation experiment and results analysis of the method proposed in this paper, which proves the practicability, effectiveness, and security of the method proposed in this paper. Finally, we summarize the content of the full text and put forward the prospect for the existing problems in the current research.

## 2. Basic of Risk Detection Based on RRT

No one can deny that risk detection of autonomous vehicle is crucial for safe driving and, in order to realize this target, it is necessary to tackle the key problem of path planning. This section concentrates on discussing the basic theory of risk detection in autonomous vehicle and introduces the basic train of thought of RRT and AV-RRT, specifically aiming at providing theoretical basis for the research. Meanwhile, when carrying out path planning, many factors should be taken into consideration. For instance, due to the higher cost of repairing automobile caused by the length of car body, it is necessary to take the length of car body into consideration. Given that it is impossible for the target automobile to turn around on the spot, this factor should also be held in high esteem.

Overall path planning of autonomous vehicles can be deemed as path planning in two-dimensional space, which has been widely researched by scholars. At present, the path planning algorithm of automobiles mainly derives from the rectification of a traditional algorithm or mixing various algorithms together so as to meet the requirements of an efficient algorithm, length of path, and number of nodes in the path in different scenes. Rajaneesh et al. [6] managed to improve the efficiency through compressing a map, and put forwards a new algorithm for an autonomous vehicle based on Ackerman's theory. In this algorithm, RRT is replaced by Dubins [7] curve which enables it to take a sample with high efficiency under the constraint condition of the bicycle mode. However, due

to neglecting the size of the car body, driving according to path planning based on this algorithm faces a higher risk of a crash. In addition, replacing RRT with a Dubins curve also increases the length of the path. Dibyendu et al. [8–10] added the constraint condition of maximum steering angle to process the samples which operated based on RRT, which is used to take samples when the nodes do not meet the standard of maximum steering angle, and it makes path planning more reasonable. However, given that the maximum steering angle is nearly 40 degrees in general, the probability of successfully taking samples based on this algorithm is lower.

To sum up, current research on method of path planning of autonomous vehicle in two-dimensional space are not only based on a reduction in algorithm efficiency, but also have less practical value because they neglect the size of the target automobile and the direction of the headstock. As a result, this paper put forwards a new method of path planning for autonomous vehicles, aiming at improving practicability of path planning and maintaining a higher efficiency of algorithm.

The RRT algorithm, developed by Professor La Valle [11] in 1988, is used for exploring space and path planning rapidly. This algorithm has a feature of probability completeness. Using this algorithm, users are able to find the path in space no matter whether there is an obstacle or not. The basic thought [12] of this algorithm is to generate a RRT after deeming an initial node as the root node. This tree might cover the space rapidly, and an effective path can be planned when the leaf node and target node coincide or the distance between them is less than the specific threshold value. Figure 1 shows the basic theory of the RRT algorithm in two-dimensional space, and the steps are shown as follows:

(1)　Define the initial point ($q_{start}$) and target point ($q_{goal}$), and deem the later one as the root node of RRT in the space Z.
(2)　Select a sampling point $q_{rand}$ randomly.
(3)　Check whether $q_{rand}$ is in the space with obstacle, desert it while ensuring it is in the space, and then repeat last step.
(4)　Try to generate a branch $[q_{near}, q_{new}]$ along the direction of $[q_{near}, q_{rand}]$ with step space $\mathcal{E}$.
(5)　Check whether the new branch passes through an obstacle, add $q_{new}$ into RRT as a new leaf node when the new branch does not pass through an obstacle. Otherwise, repeat Step (2).
(6)　Check whether the new branch meets the requirement of being less than the threshold. If it meets the requirement, the path planning ends.
(7)　Repeat Step (2) to Step (6); if iterations exceed definitive maximum iterations, path planning can be deemed as a failure.

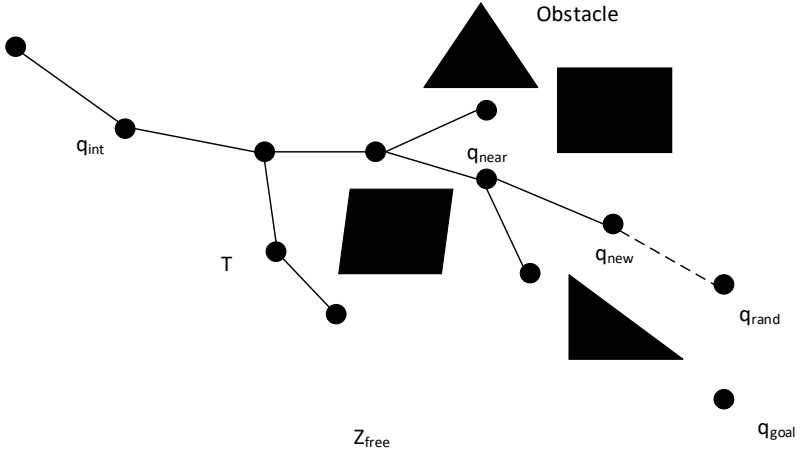

**Figure 1.** Basic RRT algorithm random expansion tree growth diagram.

It is worth mentioning that prescribing a limit to maximum iterations is to avoid falling into the condition of an infinite loop when there is no solution based on the RRT algorithm. In addition, prescribing a limit to the execution time is also conducive to solving this problem.

The RRT algorithm random spanning tree growth process is actually a process in which the change detection in the spatial environment enters the generation stage of the difference graph, which is necessary to obtain more accurate path planning [13]. The production of a rough difference map is often to extract, analyze, and calculate the relevant features affected by mathematical algorithms, and then to obtain the required feature points. In the traditional RRT algorithm, a differential comparison method is usually used for identification, so it is called a differential graph. At present, two methods are mainly used in the generation of difference chart, one is the logarithmic ratio method, and the other is the absolute difference method. The logarithmic ratio method is the analysis through the comparison of two branches. However, the biggest disadvantage of this method is that the accuracy is not high. The absolute difference method uses the method of one-to-one comparison to analyze all the known information of the branches, which is relatively more accurate but inefficient, as it is also affected by the information of the branch itself. The traditional RRT algorithm has several drawbacks in autonomous vehicles, as follows:

(1) The size of the vehicle itself and the actual situation of the vehicle in operation are ignored.
(2) In the case of dense obstacles, the complexity of this algorithm will increase significantly, resulting in low efficiency and even a low success rate.
(3) The calculation efficiency of random number will decrease when it is used for path planning in the spatial environment. If the expansion branch is added, it will not only further reduce the efficiency, but may also cause the collision of the planned path.

In order to solve the problem of the RRT algorithm when it is applied to the path planning of autonomous vehicles, the AV-RRT algorithm proposed in this paper will improve the RRT algorithm from three aspects, namely environment modeling, selection of sampling points, and growth step size of random tree leaf nodes [14], so as to improve the efficiency under the premise of the same processor and without affecting the algorithm's obstacle avoidance performance. The steps of AV-RRT are shown as follows:

Expand the pace within obstacles and make the size of the space half of the breadth of the target automobile, aiming at keeping it away from obstacles at a safe distance in practical driving. During the process of taking samples randomly, categorize samples according to the direction of headstock and maximum steering angle of the target automobile, and rectify the direction of samples which do not accord with the kinematical constraint so as to improve the rationality of path planning.

In order to reduce the time of taking valid samples in the space within the narrow path, the time for taking samples is counted, and when the number of failures reaches the threshold value, random samples are replaced by valid samples. In addition, this paper also considers the step size of RRT growth based on the principle of rapid growth, stable rise, and fast recovery, to explore the path by small step size in a space glutted with obstacles, and to explore the path by big step size in an open space, so as to improve the efficiency of the algorithm.

A flow chart of this algorithm is shown in Figure 2. In particular, $\alpha$ is defined as a number between 0 to 1, K represents maximum round of taking samples, and $\sigma$ represents maximum threshold value of distance between the target node and $q_{goal}$. Above all, when the distance between $q_{goal}$ and the nearest leaf node is less than $\sigma$ and there is no obstacle between them, $q_{goal}$ is added to RRT, and this indicates successful path planning.

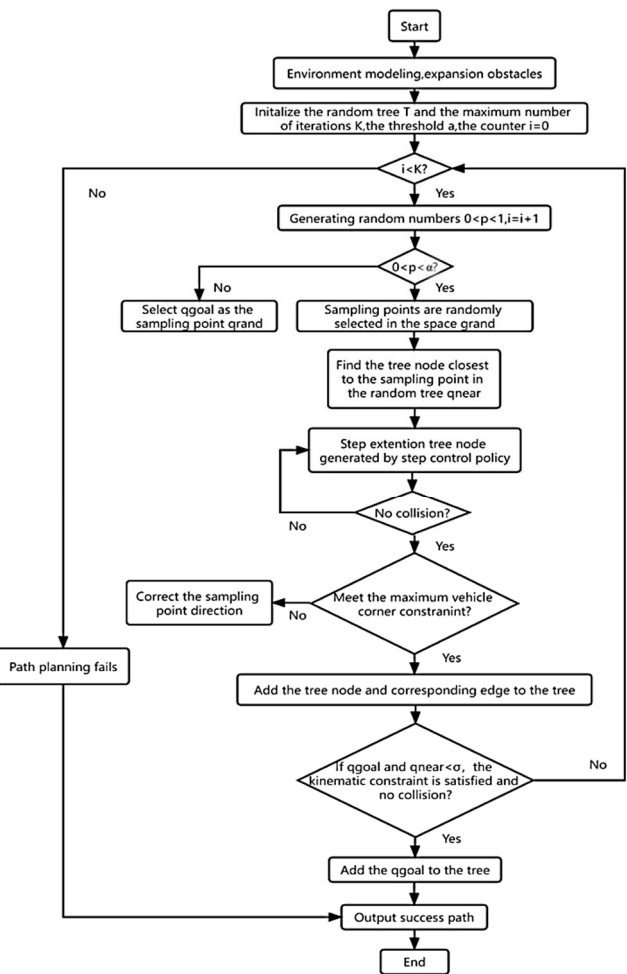

**Figure 2.** Flow Chart of the AV-RRT algorithm.

## 3. Path Detection Based on Epirelief Curve Model

In order to simplify the model, some assumptions are made, as follows:

(1) The road is assumed in the bottom half of the image.
(2) The geometrical model applied for the design of the left lane line and right lane line can be deemed a complicated curvilinear function.
(3) In the epirelief curve, the part on the left of the maximum value is the left lane line, and the part on the right of the maximum value is the right lane line.

Based on assumptions mentioned above, the black lines represent the lane edge lines, the green lines represent the maximum value on the left, and the red lines represent the maximum value on the right. The road model is shown in Figure 3.

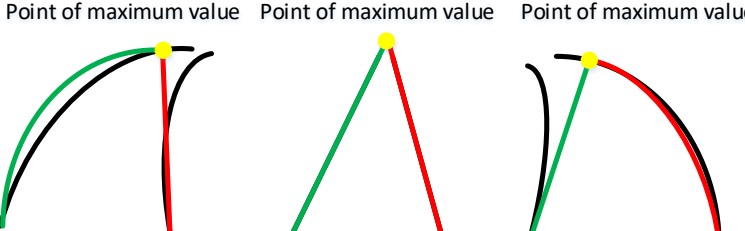

**Figure 3.** Road model.

In the road image, because the gray value at the junction of the gray pavement and the lane line is discontinuous, the pixels with abrupt changes can be extracted and applied to the edge detection of the lane line. Generally speaking, there are many kinds of detectors, such as the Sobel detector, Prewitt detector, and Connie detector. Using the angle between the lane line and the horizontal line, two kinds of direction detectors are selected in this paper. The convolution template is used to convolute the image to obtain the matrix, and the image gradient is obtained according to the peak value in the matrix to detect the edge. The calculation results and edge detection effects of different detectors are shown in Figures 4 and 5.

| -1 | -1 | 4 |
|---|---|---|
| -1 | 4 | -1 |
| 4 | -1 | -1 |

(**a**)

| 4 | -1 | -1 |
|---|---|---|
| -1 | 4 | -1 |
| -1 | -1 | 4 |

(**b**)

**Figure 4.** Detection results of two direction detectors. (**a**) 45″ lane line detector, and (**b**) 135″ lane line detector.

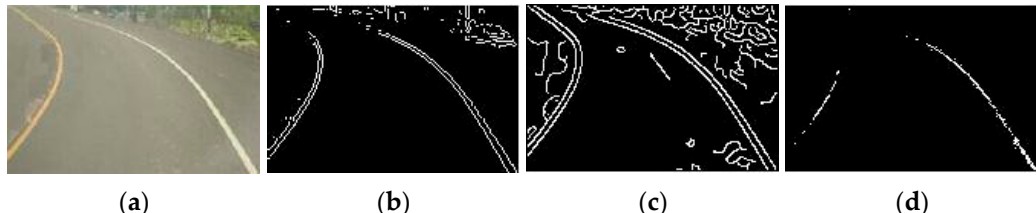

(**a**)       (**b**)       (**c**)       (**d**)

**Figure 5.** Test results of different detectors. (**a**) Original image, (**b**) Sobel detector, (**c**) Prewitt detector, and (**d**) Canny detector.

Assuming that a set of numbers $(x_0, y_0)$, $(x_1, y_1)$, ..., $(x_n, y_n)$ are the coordinates of the left and right lane lines, and it meets the condition of primitive $g(x)$, the target of using this method is to find the curve consists of minimum sums of deviation square between the observed value $y_i$ and $g(x_i)$. Through working out partial derivatives of parameters $a_0, a_1, \ldots, a_i, \ldots, a_n$ based on (1), the required parameters of the N-degree curve can be figured out, and are shown in following matrix (2). In order to make the model further accord with reality, this paper defines n as 3 in (3) according to relevant research.

$$g(x) = a_0 + a_1 x^2 + a_2 x^3 + \ldots + a_n x^n \tag{1}$$

$$\sum_{i=0}^{n} (y_i - g(x))^2 = \sum_{i=0}^{n} (y_i - (a_0 + a_1 x_i + a_2 x_i^2 + \ldots + a_n x_i^n))^2 \tag{2}$$

$$\begin{pmatrix} \sum_{i=0}^{n} 1 & \sum_{i=0}^{n} x_i & \cdots & \sum_{i=0}^{n} x_i^n \\ \sum_{i=0}^{n} x_i & \sum_{i=0}^{n} x_i^2 & \cdots & \sum_{i=0}^{n} x_i^{n+1} \\ \vdots & \vdots & \cdots & \vdots \\ \sum_{i=0}^{n} x_i^n & \sum_{i=0}^{n} x_i^{n+1} & \cdots & \sum_{i=0}^{n} x_i^{2n} \end{pmatrix} \begin{pmatrix} a_0 \\ a_1 \\ \vdots \\ a_n \end{pmatrix} = \begin{pmatrix} \sum_{i=0}^{n} y_i \\ \sum_{i=0}^{n} x_i y_i \\ \vdots \\ \sum_{i=0}^{n} x_i^n y_i \end{pmatrix} \tag{3}$$

This paper put forwards a new algorithm for the detection of lane lines based on an epirelief curve. This algorithm pays more attention to the detection and reconstruction of the left and right lane lines by virtue of the least square method, and it contributes to the optimization of the risk detection of autonomous vehicles.

## 4. Risk Detection of Autonomous Vehicle Based on NPCD

At present, due to the lack of a differential driving system, an automobile cannot spin, and, thus, both direction of headstock and maximum steering angle determine the driving direction of automobile, and the break angle between two consecutive paths [15] should be less than maximum steering angle. Due to neglecting this, path planning based on the current RRT [16] algorithm has more factors to be considered. By taking these factors into consideration, the new algorithm introduced in this paper is of higher practicability. Meanwhile, in order to improve the probability of successful sample taking and to improve the practicability, this paper adds the constraints of vehicle head direction and maximum turning angle to the sampling function. When the constraint of vehicle maximum turning angle is not satisfied, the direction is modified by using the random length vector of the current vehicle head direction and the direction of the resultant vector of the direction unit vector generated by the current random node.

This paper neglects initial direction of the headstock during the process of driving, since it is well known that the direction of the headstock is coincident with the path. Assuming that $N_i\left(x_{N_i}, y_{N_i}\right)$ is the nearest node to $q_{rand}$ in RRT and that $S_i\left(x_{S_i}, y_{S_i}\right)$ is $q_{new}$, that is to say, $P_i\left(x_{P_i}, y_{P_i}\right)$ with step size of $N_i\left(x_{N_i}, y_{N_i}\right)$ from $N_i\left(x_{N_i}, y_{N_i}\right)$ to $q_{rand}$, is the father node of $N_i\left(x_{N_i}, y_{N_i}\right)$. Define $\Delta P_i N_i$, where $\Delta P_i N_i = N_i - P_i$ as the path node from the peak to the father node and $\Delta N_i S_i = S_i - N_i$ serves as the path node from the secondary peak to the current node. Given the restraint condition of maximum steering angle $\varphi_{max}$, the break angle $\varnothing_i$ between two paths ought to accord with the condition which is shown in (4), as follows:

$$\varnothing_i = \cos \frac{\Delta P_i N_i^T \Delta N_i S_i}{|\Delta P_i N_i||\Delta N_i S_i|}, \ \forall S_i, \ \varnothing_i < \ \varphi_{max} \tag{4}$$

Steps of adopting this algorithm for detection are shown as follows:

(1)  Set threshold value $\alpha \in (0, 1)$

(2)  Generate random number $p \in (0,1)$; when $p \in (0, \alpha)$, take a sample $q_{rand}$ randomly in state space; otherwise take terminal point $q_{goal}$ as $q_{rand}$.

(3)  Check whether $q_{rand}$ is in the space within obstacles. if it is in the space, desert it and repeat Step (2).

(4)  Figure out the break angle between the two paths and check whether it accords with the constrain condition of maximum steering angle; if it does not accord, rectify the direction of the sample, or execute Step (5).

(5)  Try to work out a new branch $\left[q_{near}, q_{new}\right]$ with a step size of L and along the direction of $\left[q_{near}, q_{rand}\right]$. In particular, $q_{near}$ is the nearest node to $q_{rand}$, where $q_{new}$ is located in the line segment between $q_{near}$ and $q_{rand}$, and the distance between $q_{new}$ and $q_{near}$ is L.

(6)  Check whether the new branch passes through obstacle, add $q_{new}$ into RRT as a new leaf node when the new branch does not pass through an obstacle; otherwise, repeat Step (2).

(7)  Check whether the distance between $q_{new}$ and $q_{goal}$ is less than the threshold value and whether the new branch collides with the obstacle; if the distance is less than threshold value and new branch does not collide with obstacle, path planning can be deemed a success.

(8)  Repeat Steps (2) to (8); if number of repetitions exceed maximum iterations, path planning can be deemed a failure.

From the above AV-RRT algorithm steps, we can see that environment awareness is always the basis of path planning. The change detection process of space environment simply goes through two steps, one of which is random selection, and the other is to determine a random tree based on random selection. Each random tree can be used for independent data collection, analysis, and verification, so that it can be calculated one by one in the calculation process. At the same time, the above constraints are used to

improve efficiency and avoid infinite iteration. From Figure 6, it can be seen that point B is assumed as the branch node, which is nearest to new sample, and point A is the father node of B, while point C is located in vector $\overrightarrow{AB}$, point D is a point in the direction of the random sampling point $q_{rand}$ in the state space, and the length of $|\overrightarrow{BD}|$ is recorded as a unit vector of 1, the length of $|\overrightarrow{BC}|$ is t times $|\overrightarrow{BD}|$, $\overrightarrow{BE}$ is the resultant vector, and the included angle β between $\overrightarrow{BC}$ and $\overrightarrow{BD}$ can be calculated using Equation (5). In order to facilitate calculation, this paper assumes the coordinate of B as (0,0), and, thus, $\overrightarrow{BD} = (\cos β, \sin β)$, $\overrightarrow{BC} = (t, 0)$ and $\overrightarrow{BE} = (t + \cos β, \sin β)$. Assuming the included angle between $\overrightarrow{BC}$ and $\overrightarrow{BE}$ as γ, when γ is equal to the maximum steering angle $\varphi_{max}$, $\overrightarrow{BC} \cdot \overrightarrow{BE} = |\overrightarrow{BC}||\overrightarrow{BE}| \cos \varphi_{max}$. Thus, a conclusion as shown in (5) can be drawn.

$$t = \frac{\sin β}{\tan \varphi_{max}} - \cos β \tag{5}$$

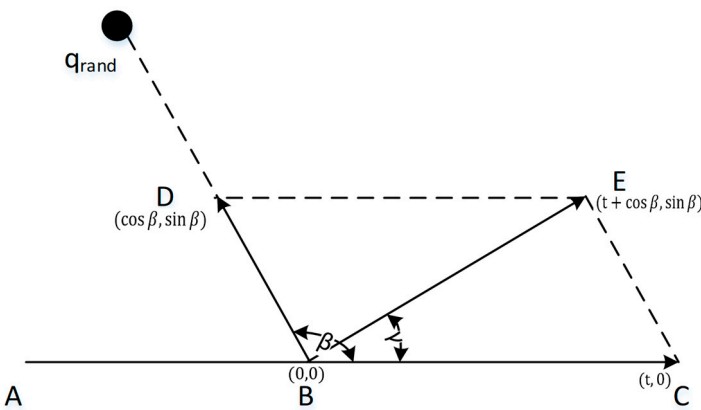

**Figure 6.** Vector diagram of path correction principle.

To be precise, the author generates a random number m ∈ [t, t + 10] and assumes it is the step size of directional vector along the direction of headstock. Then, the algorithm generates a unit vector along the direction between $q_{near}$ and $q_{rand}$. The resultant vector of these two vectors is calculated, and we regard its direction as the growth of new leaf node. Therefore, applying this algorithm might not only give impetus to taking samples successfully, but could also improve the efficiency of algorithm.

*4.1. Construction of Automobile Turning Model*

The author manages to plan an overall path by taking the size of an automobile and direction of headstock into consideration, and it seems to be an available path. When the automobile changes direction, the automobile cannot spin due to the lack of a differential driving system. In this case, given the risk of collision, which is caused by the trace of tire drafts off the safe driving zone, this paper applies NPCD [17] to the risk detection for autonomous vehicle, aiming at avoiding the risk of collision.

Furthermore, in light of the difference in driving direction and turning direction on every corner, the inside rear wheel [18] and boundary [19] lines of inside cavity are different as well. Meanwhile, there are a total of eight positional relationships according to the combination of different slopes based on the PCD algorithm [20] which might be used for detection of collision when the relationship between turning point $B(x_n, y_n)$, and its father path point $A(x_{n-1}, y_{n-1})$ as well as subsequent path point $C(x_{n+1}, y_{n+1})$ accords to any condition shown in Table 1.

**Table 1.** First conditions for detection based on NPCD.

| Number | Condition |
|--------|-----------|
| 4.1 | $x_{n-1} = x_n$ and $\frac{y_{n+1}-y_n}{x_{n+1}-x_n} > 0$ |
| 4.2 | $x_{n+1} = x_n$ and $\frac{y_n-y_{n-1}}{x_n-x_{n-1}} < 0$ |
| 4.3 | $x_{n-1} \neq x_n \neq x_{n+1}$ and $0 < \frac{y_n-y_{n-1}}{x_n-x_{n-1}} < \frac{y_{n+1}-y_n}{x_{n+1}-x_n}$ |
| 4.4 | $x_{n-1} \neq x_n \neq x_{n+1}$ and $\frac{y_n-y_{n-1}}{x_n-x_{n-1}} > 0 > \frac{y_{n+1}-y_n}{x_{n+1}-x_n}$ |

When the vehicle turns at the current turning point with the current path angle as the front wheel turning angle, the coordinates of the front wheel turning position point ($A_1$) inside the vehicle and the rear wheel position ($B_1$) inside the vehicle are calculated as follows:

$$\begin{cases} x_{A_1} = x_n + \dfrac{a_1^2 w}{2\sqrt{a_1^2+b_1^2}} \\ y_{A_1} = y_n + \dfrac{b_1 w}{2\sqrt{a_1^2+b_1^2}} \end{cases} \tag{6}$$

$$\begin{cases} x_{B_1} = x_n - \dfrac{a_1 l}{\sqrt{a_1^2+b_1^2}} \\ y_{B_1} = y_n - \dfrac{b_1 l}{\sqrt{a_1^2+b_1^2}} \end{cases} \tag{7}$$

When extending the path for W/2 before turning along the direction of center of swerve, the formula of the inside cavity boundary line ($l_1$) is shown as follows:

$$a_1 x + b_1 y + c_1 - \frac{w}{2}\sqrt{a_1^2 + b_1^2} = 0 \tag{8}$$

When extending the path for W/2 after turning along the direction of center of swerve, the formula of the inside cavity boundary line ($l_2$) is shown as follows:

$$a_2 x + b_2 y + c_2 - \frac{w}{2}\sqrt{a_2^2 + b_2^2} = 0 \tag{9}$$

When the front wheel starts to swerve, the formula of the vertical line which passes through the trace line of the inside front wheel is shown as follows:

$$b_2 x - a_2 y + a_2 y_n - b_2 x_n = 0 \tag{10}$$

When the front wheel starts to swerve, the formula of the vertical line which passes through the trace line of the inside rear wheel is shown as follows:

$$b_1 x - a_1 y + a_1 y_n - b_1 x_n + l\sqrt{a_1^2 + b_1^2} = 0 \tag{11}$$

According to the basic theory of Ackerman, a conclusion can be drawn that the traces of four wheels are concentric circles; combined with (10) and (11), the coordinate of the center of turning (O′), on the condition that the front wheel of the automobile turns for the brake angle between two path segments, can be calculated as follows:

$$\begin{cases} x_{o'} = x_n + \dfrac{la_2\sqrt{a_1^2+b_1^2}}{a_1 b_2 - a_2 b_1} \\ y_{o'} = y_n + \dfrac{lb_2\sqrt{a_1^2+b_1^2}}{a_1 b_2 - a_2 b_1} \end{cases} \tag{12}$$

In order to prevent collision during vehicle turning, the NPCD collision detection algorithm ignores the deviation between the track line of tire motion and the boundary line of the cavity. This algorithm can extract and calculate the feature information of

spatial objects on the basis of AV-RRT, and transfer and store it on the network, effectively improving the efficiency and accuracy of the algorithm. However, the following three conditions must be strictly met in the implementation process:

(1)    The boundary line inside can be calculated;
(2)    The turning center is in a straight line with the rear wheel turning center;
(3)    The turning line in the rear wheel is the track line. According to the above constraints, the following constraint equations can be determined:

$$\begin{cases} x_n - \frac{a_1 l}{\sqrt{a_1^2 + b_1^2}} \leq x_{ki} \leq x_n + \frac{a_1 w}{2\sqrt{a_1^2 + b_1^2}} \\ y_n - \frac{b_1 l}{\sqrt{a_1^2 + b_1^2}} \leq y_{ki} \leq y_n + \frac{b_1 w}{2\sqrt{a_1^2 + b_1^2}} \\ a_1 x_{ki} + b_1 y_{ki} > c_1 + \frac{w}{2}\sqrt{a_1^2 + b_1^2} \\ a_2 x_{ki} + b_2 y_{ki} < c_2 + \frac{w}{2}\sqrt{a_2^2 + b_2^2} \\ (x_{ki} - x_{o'})^2 + (y_{ki} - y_{o'})^2 > r_4^2 \end{cases} \tag{13}$$

If any peak or side of a polygon consists of obstacles which are in the closed region, the automobile will collide with the obstacle and fail to reach its destination safely.

### 4.2. Optimization of Risk Detection for Autonomous Vehicle

It is also assumed that the turning point of the vehicle is B, the turning point is A, and the turning point is C. In order to prevent collision, the following conditions as shown in Table 2 should be met.

**Table 2.** Second conditions for detection based on NPCD.

| Number | Condition |
|--------|-----------|
| 4.5 | $x_{n-1} = x_n$ and $\frac{y_{n+1} - y_n}{x_{n+1} - x_n} < 0$ |
| 4.6 | $x_{n+1} = x_n$ and $\frac{y_n - y_{n-1}}{x_n - x_{n-1}} > 0$ |
| 4.7 | $x_{n-1} \neq x_n \neq x_{n+1}$ and $\frac{y_n - y_{n-1}}{x_n - x_{n-1}} > \frac{y_{n+1} - y_n}{x_{n+1} - x_n} > 0$ |
| 4.8 | $x_{n-1} \neq x_n \neq x_{n+1}$ and $\frac{y_n - y_{n-1}}{x_n - x_{n-1}} < 0 < \frac{y_{n+1} - y_n}{x_{n+1} - x_n}$ |

Set the front inner vehicle position as ($A_1$) and the rear outer vehicle position as ($B_1$). The coordinates can be expressed as follows:

$$\begin{cases} x_{A_1} = x_n - \frac{a_1 w}{2\sqrt{a_1^2 + b_1^2}} \\ y_{A_1} = y_n - \frac{b_1 w}{2\sqrt{a_1^2 + b_1^2}} \end{cases} \tag{14}$$

$$\begin{cases} x_{B_1} = x_n + \frac{a_1 l}{\sqrt{a_1^2 + b_1^2}} \\ y_{B_1} = y_n + \frac{b_1 l}{\sqrt{a_1^2 + b_1^2}} \end{cases} \tag{15}$$

According to the previously set conditions, two sets of boundary line equations can be obtained as follows:

$$a_1 x + b_1 y + c_1 + \frac{w}{2}\sqrt{a_1^2 + b_1^2} = 0 \tag{16}$$

$$a_2 x + b_2 y + c_2 + \frac{w}{2}\sqrt{a_2^2 + b_2^2} = 0 \tag{17}$$

According to Formula (10), the formula of vertical line passing through the track line inside the rear wheel is as follows:

$$b_1 x - a_1 y + a_1 y_n - b_1 x_n - l\sqrt{a_1^2 + b_1^2} = 0 \tag{18}$$

By simultaneous use of Formulas (10) and (18), the turning center coordinates can be calculated as follows:

$$\begin{cases} x_{o'} = x_n - \dfrac{la_2\sqrt{a_1^2 + b_1^2}}{a_1 b_2 - a_2 b_1} \\ y_{o'} = y_n - \dfrac{lb_2\sqrt{a_1^2 + b_1^2}}{a_1 b_2 - a_2 b_1} \end{cases} \tag{19}$$

Similarly, the deviation between the wheel path and the boundary line of the cavity where $l_2$ is located will be ignored at the moment of turning completion. Therefore, according to the above three constraints, the following can be concluded:

$$\begin{cases} x_n + \dfrac{a_1 l}{\sqrt{a_1^2 + b_1^2}} \ \leq\ x_{ki} \ \leq\ x_n - \dfrac{a_1 w}{2\sqrt{a_1^2 + b_1^2}} \\ y_n + \dfrac{b_1 l}{\sqrt{a_1^2 + b_1^2}} \ \leq\ y_{ki} \ \leq\ y_n - \dfrac{b_1 w}{2\sqrt{a_1^2 + b_1^2}} \\ a_1 x_{ki} + b_1 y_{ki} \ <\ c_1 + \dfrac{w}{2}\sqrt{a_1^2 + b_1^2} \\ a_2 x_{ki} + b_2 y_{ki} \ >\ c_2 + \dfrac{w}{2}\sqrt{a_2^2 + b_2^2} \\ (x_{ki} - x_0)^2 + (y_{ki} - y_0)^2 \ >\ r_4^2 \end{cases} \tag{20}$$

As in the first condition, as long as any peak or side of polygon which consists of obstacles is in the above constraint area, the vehicle will collide when turning.

### 4.3. PM Path Modification Strategy

The author has attempted to apply NPCD [21] into the collision detection of an automobile on every corner based on its feature of turning, and it ended up with a failure. In order to save time, the author has tried to rectify the path near the corners which do not pass the detection according to the feature of minimum steering radius, aiming at discovering a new shortest path for the automobile. The reason for the detection ending in failure is due to some obstacles in the detection region. Given that the region consists of $l_1$ inside the cavity boundary line, a direct line from O′ to the position where the inside rear wheel is located on after a swerve, and the track line of turning of rear wheel, the failure can be deemed as being caused by the position of the trod line of the inside rear wheel. Meanwhile, given that the trod line of the inside front wheel coincides with the boundary line of the inside safe cavity, collision can be avoided by making the trod line of the inside rear wheel pass through the turning point of the inside front wheel. Details of the rectification strategy for path planning based on PM [22] are shown as follows:

(1)  Before the automobile reaches its turning point, the author shifts it so as to make its rear wheel located on the circumference, which consists of turning the trod line of the inside front wheel when the automobile reaches a turning point. In this case, when the automobile is turning for the break angle between two segments, its inside rear wheel is located on the turning point of the inside boundary of the safe cavity.

(2)  There is no doubt that driving in the safe cavity is safe, and, thus, prolonging the distance of driving in the safe cavity will reduce the probability of collision. In order to achieve this goal, it is necessary to implement deviation treatment and guarantee the shortest distance.

Figure 7 shows the trace of the wheel during the process of deviation. In particular, the red automobile indicates the position of turning without deviation, the black dotted automobile indicates the position of starting deviating and completing deviation, and the blue arc indicates the trod line of the rear wheel during the process of deviation. Point B is the current turning point of the vehicle, point A is the previous path point of point B on the

global path, that is, the parent path point of point B, and point C is the next path point of point B on the global path, that is, the subsequent path point of point B.

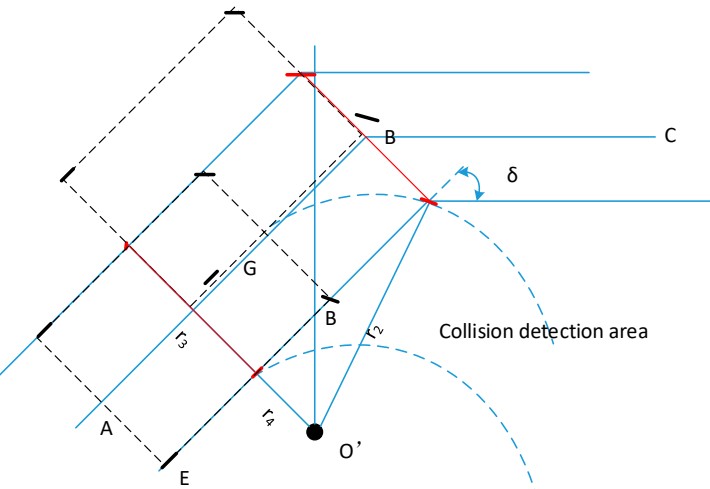

**Figure 7.** Schematic diagram of the path offset process.

When turning before reaching the turning point, the distance between the trod line of the inside front wheel and that of the inside rear wheel can be calculated as follows:

$$r = r_2 - r_4 = \frac{l}{\sin \delta} - \frac{l}{\tan \delta} \tag{21}$$

In particular, $\delta$ means the angle of turning of front wheel, and it is equal to the path $\varnothing$. In order to make the shortest path during the process of deviation, the automobile drives along the minimum turn radius, that is to say, the turning angle of the front wheel reaches its maximum ($\varphi_{\max}$), and the turn radius of inside rear wheel can be calculated as follows:

$$r_4 = \frac{l}{\tan \varphi_{\max}} \tag{22}$$

During this process, the race of each wheel can be deemed as consisting of the same two arcs. An automobile is a rigid body with a fixed size and, thus, making the position of one wheel clear is equal to making the other three wheels clear. This paper deems the original inside rear wheel to be the datum, and when the automobile shifts transversely for $\Delta r$, it ought to drive forward for $\Delta l$.

$$\Delta l = 2\sqrt{r_4^2 - \left(r_4 - \frac{1}{2}\Delta r\right)^2} \tag{23}$$

Assuming that formula of path is expressed as follows:

$$a_1 x + b_1 y + c_1 = 0 \tag{24}$$

then the overall coordination system included angle $\theta$ between formula of path and X-axis can be calculated by following formula:

$$\theta = \arctan\left(-\frac{a_1}{b_1}\right) \tag{25}$$

Corresponding to the algorithm for collision detection, this paper divides rectification of the path based on position G inside the rear wheel, which is located on the extended curve of trod line of the position of the datum wheel and the slope of the path into two conditions.

First of all, the first kind of conditions are 4.1–4.4, and are combined with the known Formulas (6) and (7). From Figure 7 it can be seen that when the datum wheel reaches point E, it starts to turn outward for $\varphi_{max}$. When reaching point F, the datum wheel starts to turn inward for $\varphi_{max}$. When reaching the side front wheel without deviation, in this case, the automobile might turn for $\varnothing$ without colliding with an obstacle, that is to say, path planning is completed. Coordinates of point E, point F, and point G are shown as follows:

$$\begin{cases} x_E = x_n - \dfrac{a_1 l}{\sqrt{a_1^2 + b_1^2}} + \Delta l \cos\theta \\ y_E = y_n - \dfrac{b_1 l}{\sqrt{a_1^2 + b_1^2}} + \Delta l \sin\theta \end{cases} \tag{26}$$

$$\begin{cases} x_F = x_n - \dfrac{a_1 l}{\sqrt{a_1^2 + b_1^2}} + \Delta l \cos\theta + \frac{1}{2}\Delta r \sin\theta \\ y_F = y_n - \dfrac{b_1 l}{\sqrt{a_1^2 + b_1^2}} + \Delta l \sin\theta - \frac{1}{2}\Delta r \cos\theta \end{cases} \tag{27}$$

$$\begin{cases} x_G = x_n - \dfrac{a_1 l}{\sqrt{a_1^2 + b_1^2}} + \Delta r \sin\theta \\ y_G = y_n - \dfrac{b_1 l}{\sqrt{a_1^2 + b_1^2}} - \Delta r \cos\theta \end{cases} \tag{28}$$

The second kind of conditions are 4.5–4.8 and, combined with the known Formulas (14) and (15), from the above information in Figure 7, the coordinates of point E, point F, and point G are shown as follows:

$$\begin{cases} x_E = x_n + \dfrac{a_1 l}{\sqrt{a_1^2 + b_1^2}} - \Delta l \cos\theta \\ y_E = y_n + \dfrac{b_1 l}{\sqrt{a_1^2 + b_1^2}} - \Delta l \sin\theta \end{cases} \tag{29}$$

$$\begin{cases} x_F = x_n + \dfrac{a_1 l}{\sqrt{a_1^2 + b_1^2}} - \Delta l \cos\theta - \frac{1}{2}\Delta r \sin\theta \\ y_F = y_n + \dfrac{b_1 l}{\sqrt{a_1^2 + b_1^2}} - \Delta l \sin\theta + \frac{1}{2}\Delta r \cos\theta \end{cases} \tag{30}$$

$$\begin{cases} x_G = x_n + \dfrac{a_1 l}{\sqrt{a_1^2 + b_1^2}} - \Delta r \sin\theta \\ y_G = y_n + \dfrac{b_1 l}{\sqrt{a_1^2 + b_1^2}} + \Delta r \cos\theta \end{cases} \tag{31}$$

This section puts forward a new algorithm for collision detection of a non-particle automobile according to the feature of turning of autonomous vehicle and a rectification strategy for path planning based on PM [23,24].

## 5. Analysis on Emulation Experiment

This section concentrates on examining the validity of the algorithm for risk detection in autonomous vehicles. This paper tries to discuss the emulation experiment from three perspectives, including experiment scene, experiment design, and experiment result.

### 5.1. Experiment Scene

The emulated experiment scene of AV-RRT is a virtual two-dimensional plane with polygonal obstacles. In order to maintain the universality of the algorithm, this paper divides the experiment into three conditions, including a two-dimensional plane with sparse obstacles, with moderate obstacles, and with dense obstacles. From Figure 8, it can be seen that closed polygons represent obstacles, and the blank region represents space in which the automobile might drive freely. In addition, the size of the scene is 500 pixels * 500 pixels, and the coordinates the of moving entity are expressed by pixels.

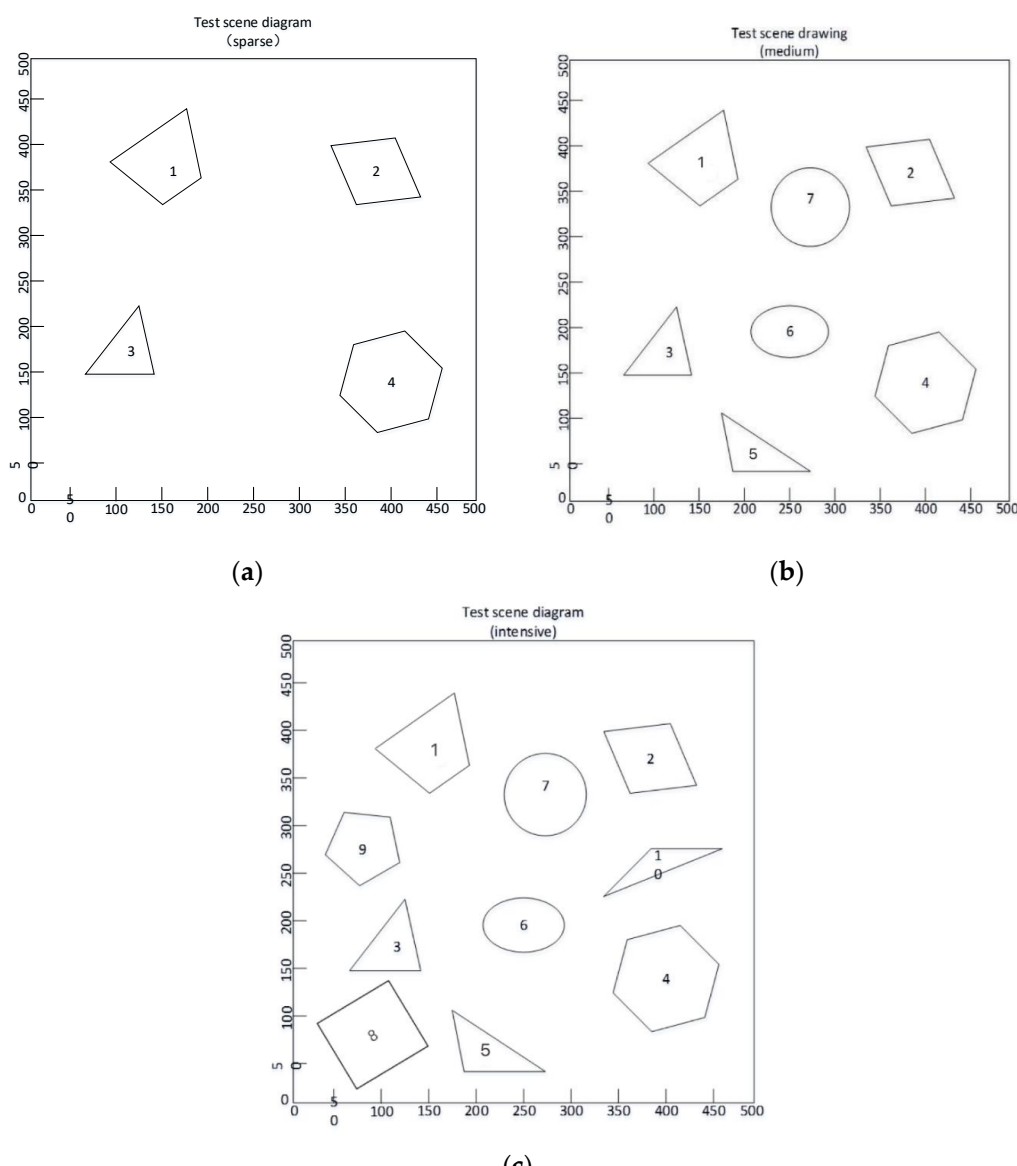

**Figure 8.** Simulation experiment scene of sparse, moderate, and dense obstacles. (**a**) Sparse obstacle simulation, (**b**) Moderate obstacle simulation and (**c**) dense obstacle simulation.

Figure 8 shows the number of obstacles and their coordinates in different scenes. And the coordinates of salient points under the plane of different scenes are shown in the Tables 3–5.

**Table 3.** Coordinates of salient points under the plane with sparse obstacles condition.

| Salient Point | 1 | 2 | 3 | 4 | 5 | 6 |
|---|---|---|---|---|---|---|
| Obstacles 1 | (169,103) | (181,42) | (272,42) | | | |
| Obstacles 2 | (332,399) | (358,329) | (404,408) | (434,336) | | |
| Obstacles 3 | (63,149) | (121,223) | (142,150) | | | |
| Obstacles 4 | (345,121) | (356,180) | (383,80) | (416,199) | (439,102) | (455,152) |

**Table 4.** Coordinates of salient points under plane with moderate obstacles condition.

| Salient Point | 1 | 2 | 3 | 4 | 5 | 6 |
|---|---|---|---|---|---|---|
| Obstacles 1 | (169,103) | (181,42) | (272,42) | | | |
| Obstacles 2 | (332,399) | (358,329) | (404,408) | (434,336) | | |
| Obstacles 3 | (63,149) | (121,223) | (142,150) | | | |
| Obstacles 4 | (345,121) | (356,180) | (383,80) | (416,199) | (439,102) | (455,152) |
| Obstacles 5 | (87,379) | (145,333) | (177,438) | (189,361) | | |
| Obstacles 6 | (202,198) | (250,162) | (250,225) | (289,198) | | |
| Obstacles 7 | (229,332) | (265,289) | (265,376) | (312,332) | | |

**Table 5.** Coordinates of salient points under the plane with dense obstacles condition.

| Salient Point | 1 | 2 | 3 | 4 | 5 | 6 |
|---|---|---|---|---|---|---|
| Obstacles 1 | (169,103) | (181,42) | (272,42) | | | |
| Obstacles 2 | (332,399) | (358,329) | (404,408) | (434,336) | | |
| Obstacles 3 | (63,149) | (121,223) | (142,150) | | | |
| Obstacles 4 | (345,121) | (356,180) | (383,80) | (416,199) | (439,102) | (455,152) |
| Obstacles 5 | (87,379) | (145,333) | (177,438) | (189,361) | | |
| Obstacles 6 | (202,198) | (250,162) | (250,225) | (289,198) | | |
| Obstacles 7 | (229,332) | (265,289) | (265,376) | (312,332) | | |
| Obstacles 8 | (25,89) | (69,21) | (104,138) | (148,66) | | |
| Obstacles 9 | (38,272) | (59,318) | (71,238) | (111,307) | (118,260) | |
| Obstacles 10 | (332,224) | (383,273) | (457,273) | | | |

Owing to applying pixel map into AV-RRT, the universality of this algorithm has improved a lot.

### 5.2. Design of Emulation Experiment

This paper divides the emulation experiment into two parts, including functional examination and performance examination.

In terms of functional examination, the main advantage of AV-RRT lies in taking the size of automobile and direction of the headstock into consideration and adopting dynamic step size.

In terms of performance examination, this paper tries to compare performance in terms of time consumption [25,26], times of node extension and number of nodes based on AV-RRT based on RRT, and performance in path planning in different scenes based on AV-RRT based on RRT. In order to avoid the influence of initial position and targeting position, this paper defines them randomly and carries out path planning for 400 iterations.

### 5.3. Analysis on Result of Emulation Experiment

On the premise of not losing generality, this experiment should use the verification experiment shown in Figure 8b. In the simulation experiment, in order to better observe the experiment, the obstacles in the figure are expanded without data deviation. Therefore, the following basic conditions are set: the length of the vehicle is 4 m, the width is 1.8 m, the maximum turning angle is 40 degrees, and the expansion reaction size of the obstacle is 0.9 m. Figure 9 shows the map after swelling.

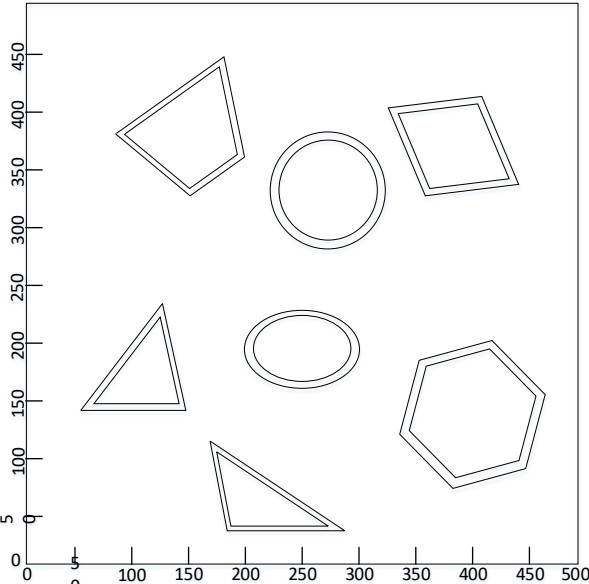

**Figure 9.** Map processed with the swelling treatment.

This paper assumes that the coordinates of the initial point are (1,1), and that the targeting point is at (499,499). Under this circumstance, this paper implements experiment and comparison by virtue of controlling variables.

The RRT algorithm is used for calculation. As shown in Figure 10, in the simulation experiment, the path planning time is 54 s, and the number of extended nodes reaches more than 2000; the number of nodes in the return path is 252. The overall calculation efficiency is low, and the final path is not a more scientific and reasonable path.

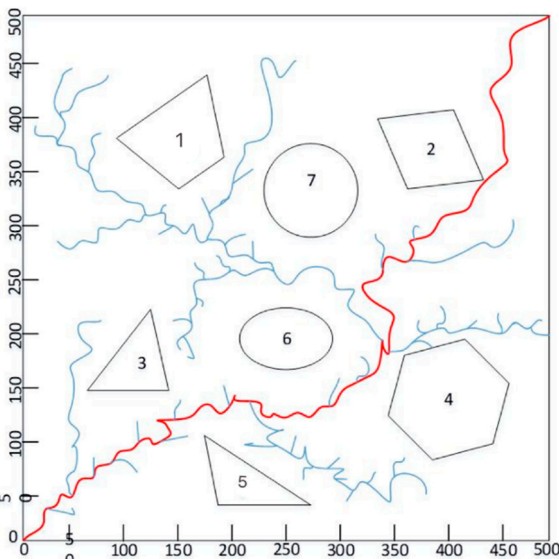

**Figure 10.** Result of path planning based on RRT.

(1)    Path planning based on AV-RRT accords to the kinematical constraint condition.

From Figure 11, it can be seen that when using the AV-RRT algorithm to pass through a curve with a turning angle of 40 degrees, the required path planning time is 44 s, and the number of expanded nodes is more than 1300, of which the maximum turning angle is 37 degrees, which means the result of path planning is ideal.

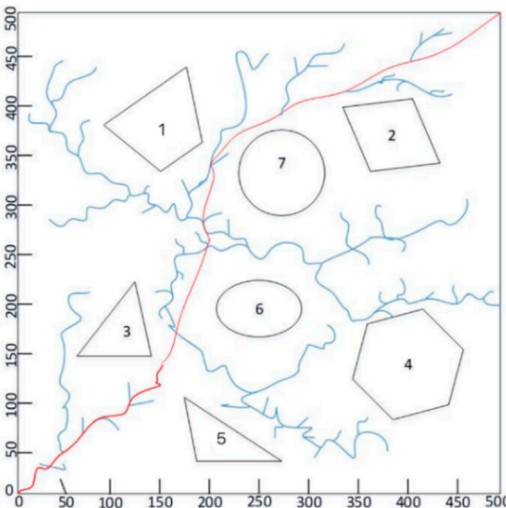

**Figure 11.** Result of path planning according with kinematical constraint condition.

(2)   Functional examination for AV-RRT algorithm

In Figure 12 it can be seen that because the control of step size is based on principle of rapid growth, the number of summary points in the simulation experiment is 454, the number of nodes in the path is 122, the path planning time is 28 s, and the entire path planning time is reduced by nearly half, which proves that the dynamic step size of the algorithm is effective.

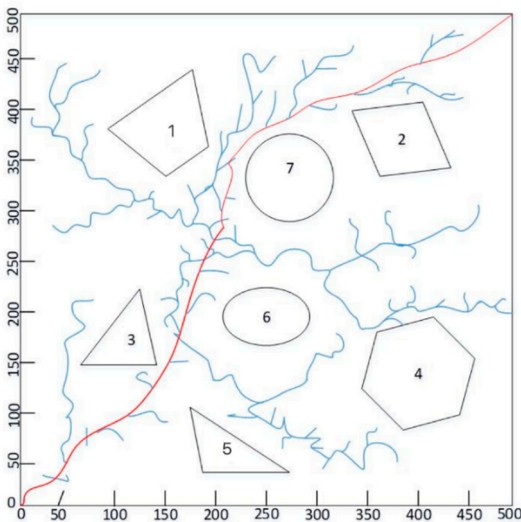

**Figure 12.** Result of path planning according to dynamic step size.

(3)   Functional Examination of Algorithm Based on AV-RRT

Figure 13 shows the result of path planning based on the AV-RRT algorithm. During the experiment, it takes 48 s to carry out path planning, the number of extended nodes in the space is 965, the number of nodes in the return path is 132, and the length of path planned is 861 m. In particular, comparing with path planning based on RRT, a conclusion can be drawn that extended nodes in the space decreased by 63%, path nodes decreased by 49%, consumption of time decreased by 11%, and length of path decreased by 17%. Furthermore, the maximum steering angle is 39.87 degrees, which appears on the position of 11th node, that is to say, the result of path planning is ideal.

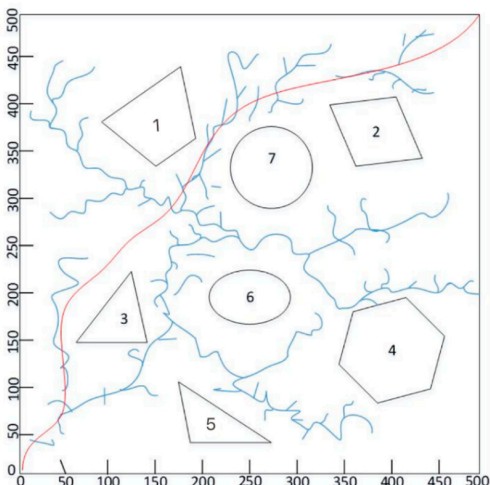

**Figure 13.** Result of path planning based on the AV-RRT algorithm.

To sum up, the result of the functional examination is ideal, and this paper will be devoted to carrying out a performance examination in the following section.

From the above simulation results, we can see that the AV-RRT algorithm is significantly better than the RRT algorithm in terms of path planning time and dynamic step size. The number of expansion nodes in space is reduced by more than 60%, the time of path planning is reduced by about 50%, and the maximum turning radius also conforms to the vehicle's kinematic constraints.

### 5.4. Analysis on Performance Examination Based on AV-RRT Algorithm

This paper defines the number of extended nodes, number of nodes of path planning, and time consumption of path planning as the key performance indexes.

In order to maintain the universality of the algorithm, this paper divides the experiment into three conditions, including a two-dimensional plane with sparse obstacles, a plane with moderate obstacles, and a plane with dense obstacles. The experiment is conducted 30 times, which paves the way for comparison. In addition, this paper selects the experiment scene randomly, and defines (0,0) and (499,499) as the initial point and final point, respectively, on the premise of having no influence on the result of the experiment.

(1)    Comparison of Extended Nodes [27] Based on AV-RRT and based on RRT

The difference between the two is shown in Figures 14–16. The blue represents the number of expansion nodes of the basic RRT algorithm, and the orange represents the number of expansion nodes of the AV-RRT algorithm.

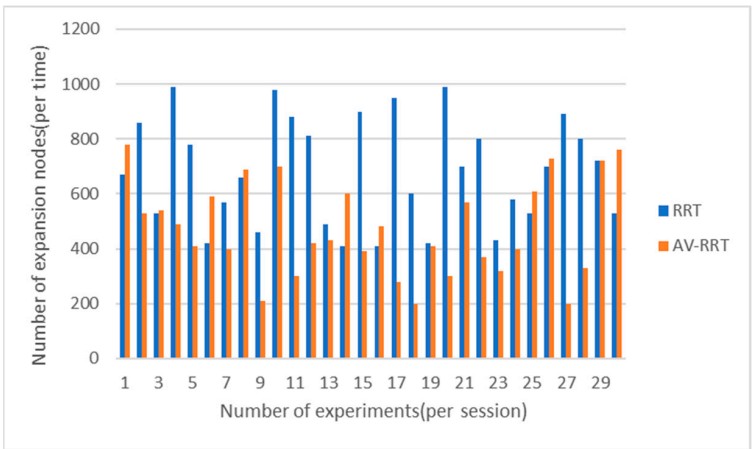

**Figure 14.** Comparison of number of extended nodes in scene with sparse obstacles.

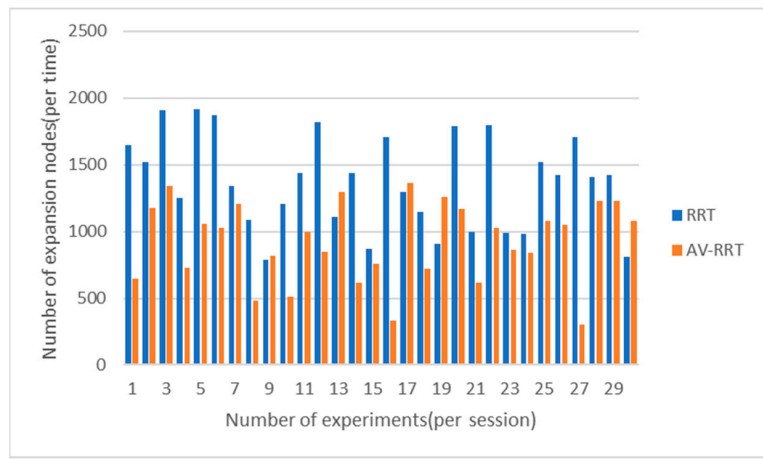

**Figure 15.** Comparison of number of extended nodes in scene with moderate obstacles.

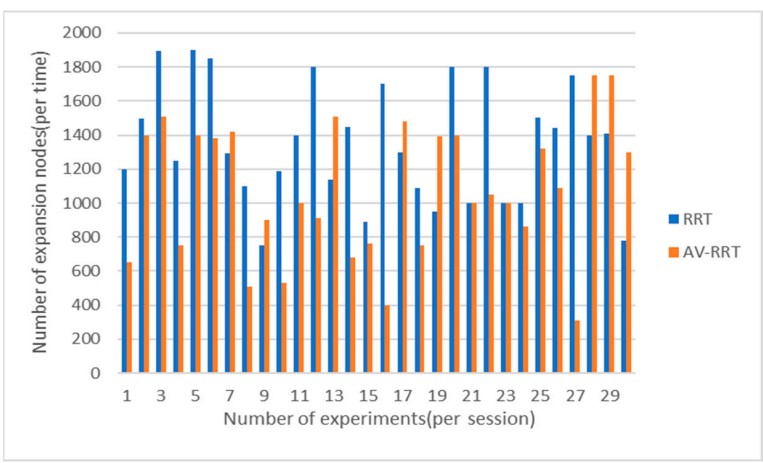

**Figure 16.** Comparison of number of extended nodes in scene with dense obstacles.

From Figures 14–17, it can be seen that in each scene, the performance is affected by random growth in the space time of extension, meaning that the number of extended nodes based on both the RRT and AV-RRT is bigger. However, the latter algorithm is more stable than the former, and the time of extension of extended nodes based on the AV-RRT is smaller compared to RRT, especially in the scene with sparse obstacles, which is caused by dynamic step size.

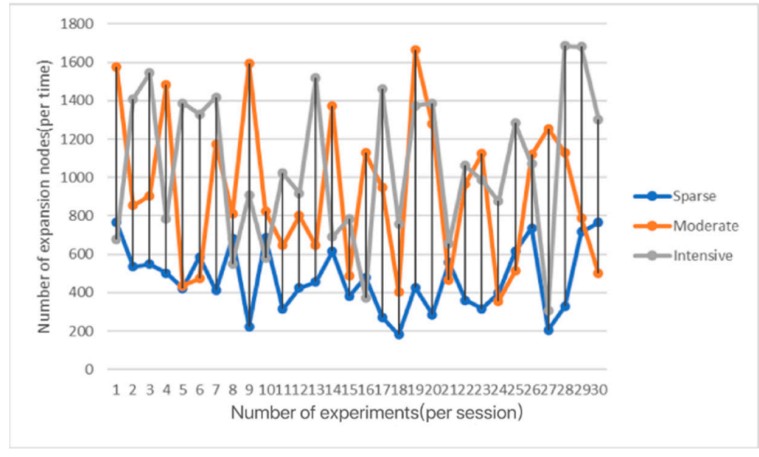

**Figure 17.** Comparison of number of extended nodes in each scenario for the AV-RRT algorithm.

(2) Comparison of Path Nodes based on AV-RRT and based on RRT

From Figures 18–21, it can be seen that in each scene, the number of path nodes based on AV-RRT is smaller than RRT, especially in the scene with sparse obstacles. This is caused by the dynamic step size.

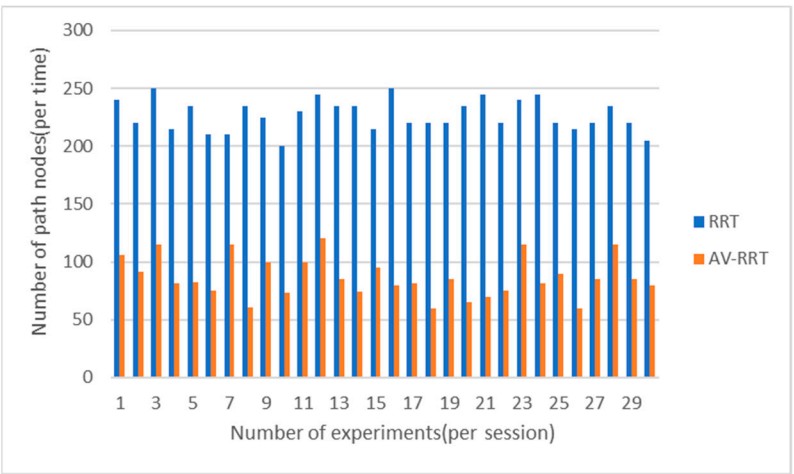

**Figure 18.** Comparison of the number of path nodes in the scene with sparse obstacles.

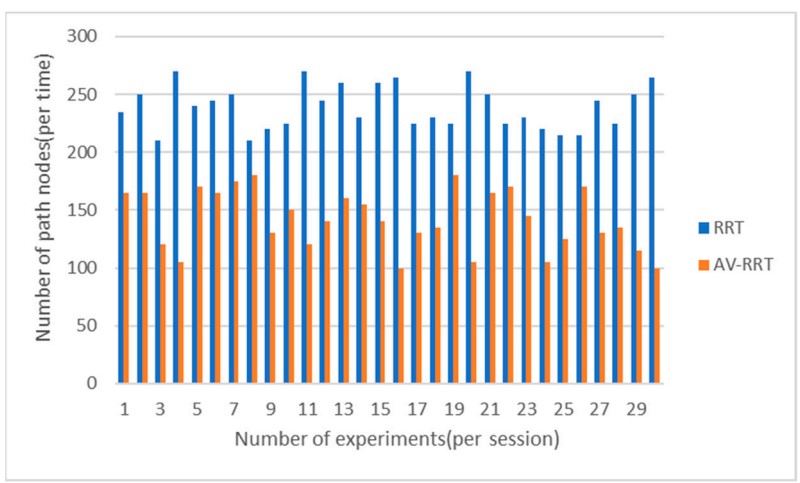

**Figure 19.** Comparison of the number of path nodes in the scene with moderate obstacles.

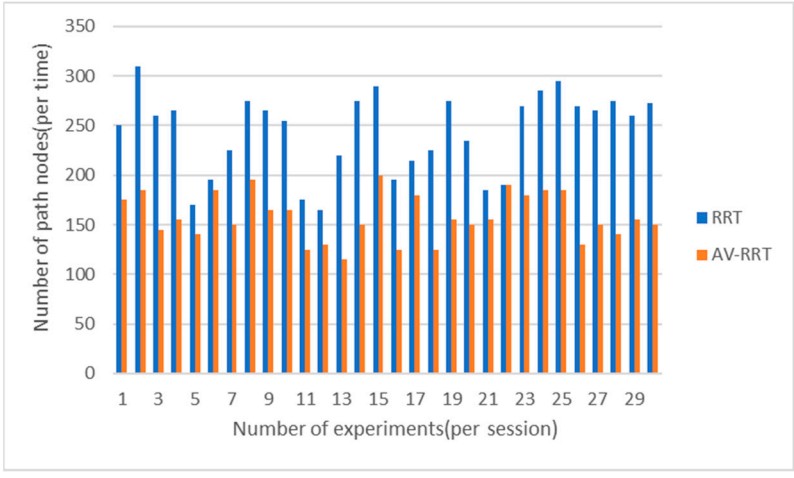

**Figure 20.** Comparison of the number of path nodes in the scene with dense obstacles.

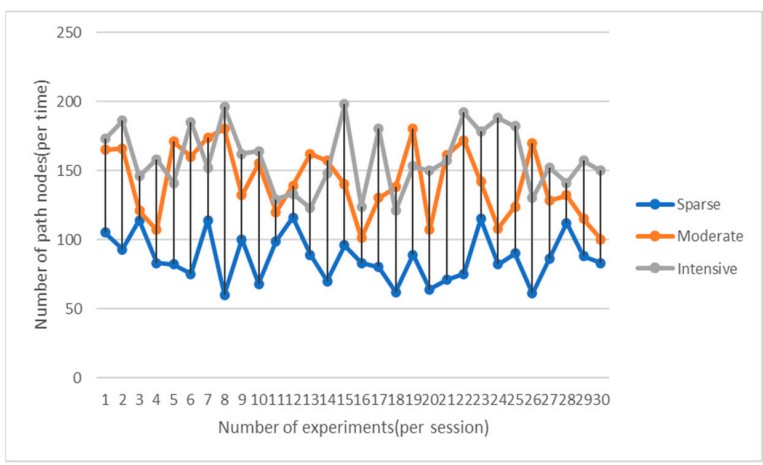

**Figure 21.** Comparison for number of return path nodes in each scenario with AV-RRT algorithm.

(3)    Comparison of Consumption of Time based on AV-RRT and based on RRT

From Figures 22–25, it can be seen that in each scene, consumption of time for both the RRT algorithm and AV-RRT algorithm is greater. However, the latter algorithm is more stable than the former, and the consumption of time by the AV-RRT algorithm is shorter than RRT algorithm, especially in the scene with sparse obstacles, which is caused by the dynamic step size.

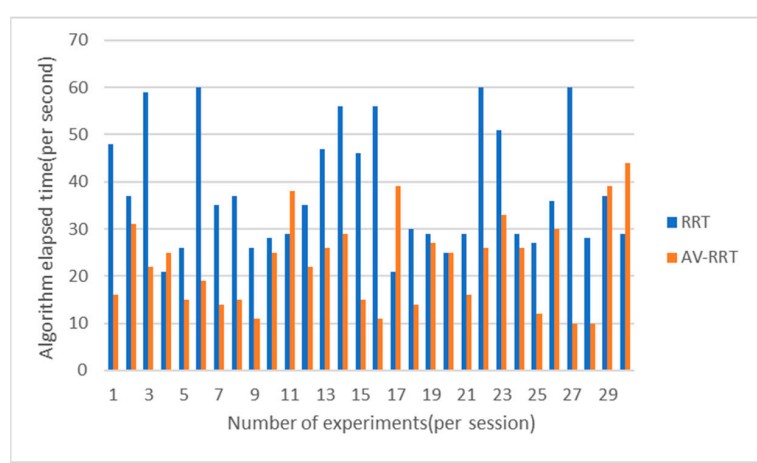

**Figure 22.** Comparison of consumption of time in the scene with sparse obstacles.

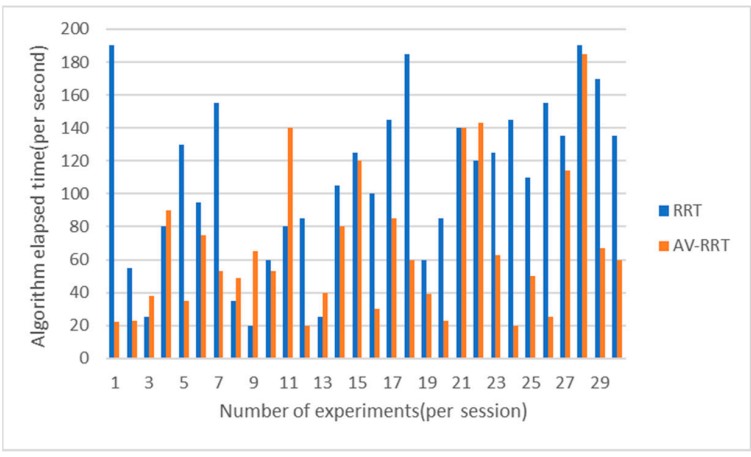

**Figure 23.** Comparison of consumption of time in the scene with moderate obstacles.

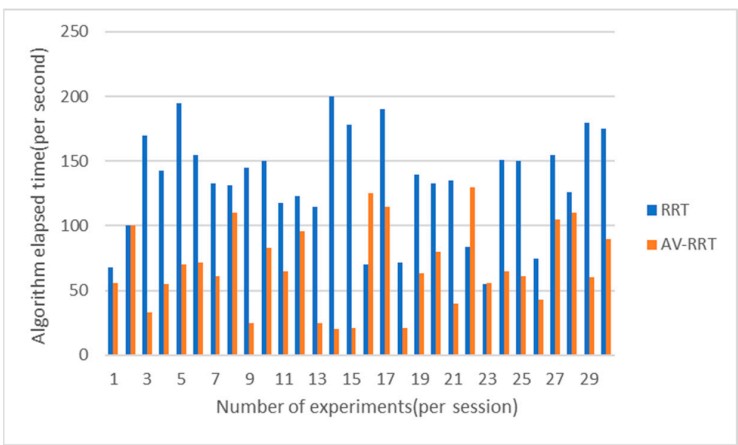

**Figure 24.** Comparison of consumption of time in the scene with dense obstacles.

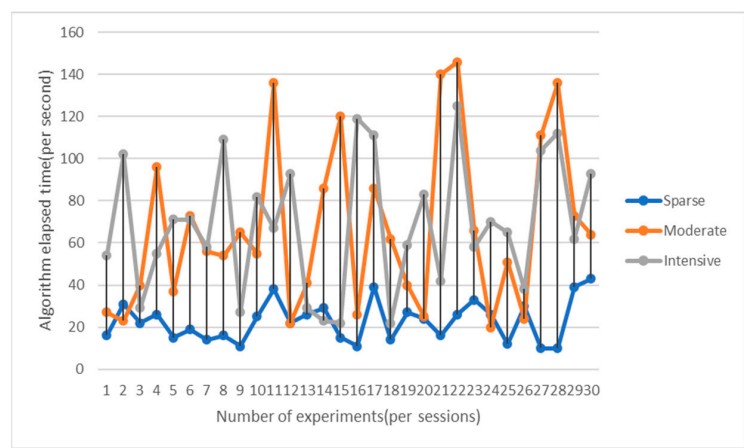

**Figure 25.** Comparison of consumption of time in each scenario with the AV-RRT algorithm.

Based on the simulation experimental data in the above three scenarios, it is concluded that the AV-RRT algorithm has a higher performance than the basic RRT algorithm in terms of node expansion times in space, number of nodes included in the planned path, and path planning time consumption, and this advantage is more prominent in sparse obstacle scenarios.

In order to maintain the validity of experiment, this paper selects the initial point and targeting point randomly and divides the experiment into three conditions, including a two-dimensional plane with sparse obstacles, a plane with moderate obstacles, and a plane with dense obstacles, and implements the experiment 400 times, which paves the way for comparison. Table 6 shows the result of experiment, including the AV-RRT algorithm, RRT algorithm [28], and KB-RRT [29] algorithm, which was used as a reference for comparison with other studies in the literature.

Generally speaking, although path planning based on AV-RRT accords to kinematical constraint condition and has better practical applicability, it weakens the capability of the path searching aspect of the algorithm and increases the execution time. In addition, according to the data in the table, owing to setting dynamic step size and area sampling, the performance indexes of the AV-RRT algorithm are better than the RRT algorithm. In the three scenes, times of extension for extended nodes based on the AV-RRT algorithm decreased by 30.35%, 31.69%, and 22.52%, respectively, and the total number decreased by 28.19%. Meanwhile, the number of path nodes based on the AV-RRT algorithm decreased by 61.56%, 40.63%, and 35.02%, respectively, and the total number decreased by 45.74%. Moreover, consumption of time based on the AV-RRT decreased by 39.56%, 38,64%, and 48.74%, respectively, and the total consumption of time decreased by 42.31%.

**Table 6.** Comparison of performance indexes of three algorithms.

| Scene | Algorithm | Number of Extended Nodes | Number of Path Nodes | Consumption of Time/s |
|-------|-----------|--------------------------|----------------------|------------------------|
| With sparse obstacles | RRT | 679.97 | 225.90 | 37.77 |
| | KB-RRT | 613.47 | 234.97 | 27.42 |
| | AV-RRT | 473.57 | 86.83 | 22.83 |
| With moderate obstacles | RRT | 1352.9 | 239.0 | 108.7 |
| | KB-RRT | 1442.1 | 232.46 | 78.04 |
| | AV-RRT | 924.17 | 141.90 | 66.7 |
| With dense obstacles | RRT | 1367.6 | 243.6 | 133.63 |
| | KB-RRT | 1521.7 | 236.72 | 88.47 |
| | AV-RRT | 1059.6 | 158.3 | 68.5 |

Compared with the KB-RRT algorithm, in sparse, moderate, and dense obstacle scenarios, the number of expansion nodes based on the AV-RRT algorithm is reduced by 22.80%, 35.91%, and 30.37%, respectively, and the total number of expansion nodes is reduced by 29.69%. Meanwhile, the number of path nodes based on the AV-RRT algorithm is reduced by 63.05%, 38.96%, and 33.13%, respectively, and the total number of path nodes is reduced by 45.05%. In addition, the time consumption based on AV-RRT was reduced by 16.74%, 14.53%, and 22.57%, respectively, and the total time consumption was reduced by 17.95%.

Compared with the RRT algorithm and KB-RRT algorithm, the AV-RRT algorithm has obvious performance advantages. No matter whether the obstacles are sparse, moderate, or dense, the number of expansion nodes decreases significantly, the dynamic step size has obvious advantages, and the path calculation time decreases significantly. The experimental results prove the effectiveness of AV-RRT algorithm and allow us to draw conclusions from the number of expansion nodes, dynamic step size, and path calculation time.

In order to facilitate analysis, this paper carries out an emulation experiment for the scene with moderate obstacles.

Figure 26 shows the result of path planning based on AV-RRT and the scene with moderate obstacles. Through detection based on NPCD [30], it can be seen that a turning point (337.57,400.1046) in the figure does not pass the detection.

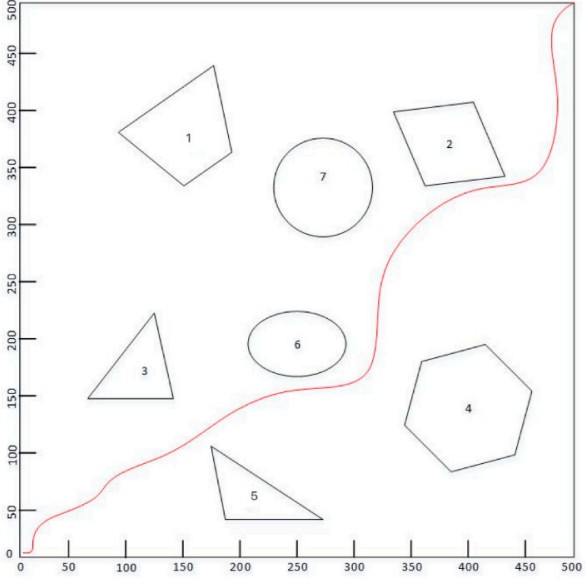

**Figure 26.** Emulation experiment of collision detection.

From Figure 27, it can be seen that when the automobile turns from the path from point (332.7231,397.5523) to point (337.57,400.1046) to the path from point (337.57,400.1046) to (346.2173,398), due to driving outside safe cavity, the inside body of the automobile collides with the obstacle. In order to avoid collision, the automobile should firstly turn for maximum steering angle, and, when completing turning, the inside rear wheel is located on the extended curve of the trod line of the inside front wheel without deviation, before it turns with the front wheel for the path steering angle.

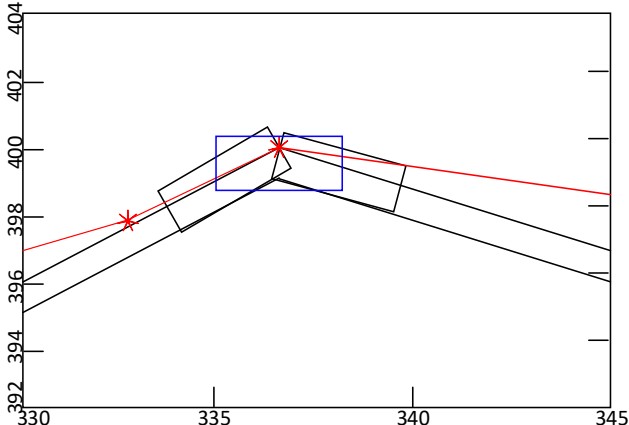

**Figure 27.** Failure of collision detection.

Due to that, the safe cavity is a region that extends the planned path based on AV-RRT to two sides for half of the breadth of the automobile; the rectification of the path based on PM lies in the original path as well. Figure 27 shows the path after rectification. The automobile turns outward on the red path point (332.8721,397.5523) for 40 degrees, and then turn to the side close to the obstacle for 40 degrees when reaching green path point (333.2901,398.5590). When the automobile reaches blue path point (333.8571,399.6457), the rectification of path planning is completed.

As shown in Figure 28, the result of the experiment mentioned above primarily proves the validity of collision detection based on NPCD and the rectification of the path based on PM. In order to further examine the validity of the algorithm, the author implemented the experiment 100 times, and the results showed that collision detection based on NPCD might forecast whether or not the automobile will collide with an obstacle, and the probability of rectifying path successfully reached 85%.

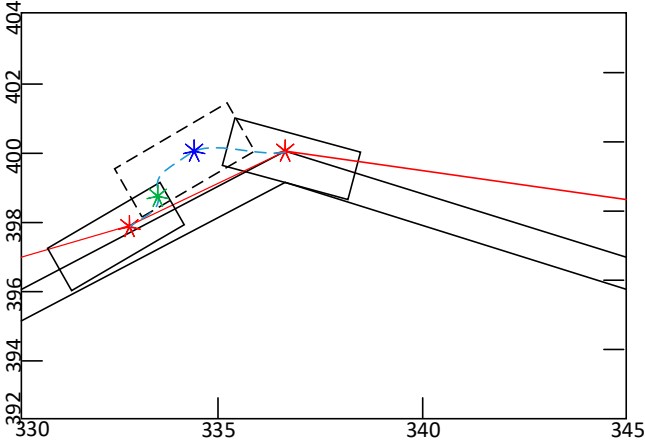

**Figure 28.** Rectification of path.

### 6. Conclusions

In this paper, an AV-RRT algorithm is proposed and implemented to solve the problems of poor practicability and low efficiency of existing path planning algorithms used in autonomous vehicles. First of all, the expansion of obstacles ensures that the vehicle can drive safely approximately along the global path drawn by the laws and regulations, and preliminarily ensures the safety of the automobile. Secondly, in the RRT algorithm sampling process, the maximum turning angle limit of the vehicle is added, the direction of the sampling points that do not meet the limit is corrected, and the fixed-point sampling method is adopted to enable the vehicle to quickly pass through the obstacle-dense area, ensuring the efficiency and success rate of the algorithm in the obstacle-dense environment. Finally, the random tree uses the dynamic step size strategy of "fast start, steady rise, and fast recovery" to grow in free space, greatly improving the efficiency of path planning. Compared with other algorithms, the superior performance of the AV-RRT algorithm is verified. At the same time, the NPCD collision detection algorithm and PM path modification strategy are proposed and implemented in this paper to avoid the risk of collision with obstacles when the vehicle is driving and turning according to the global path planned by the AV-RRT algorithm. The problem of vehicle collision detection is transformed into the problem of the relationship between the rear wheel inside the vehicle and the relative position of the obstacle, to detect whether there is a risk of collision between the vehicle and the obstacle at each turning point, and to modify the path at the turning point where the collision may occur using the PM path modification strategy. The correctness and effectiveness of the method are proved by simulation experiments, and the security of the algorithm is improved.

**Author Contributions:** Conceptualization, Y.M. and K.T.K.T.; methodology, Y.M. and K.G.L.; software Y.M.; validation, K.G.L. and K.T.K.T.; investigation Y.M.; writing—original draft preparation, Y.M.; writing—review and editing, K.G.L.; visualization, Y.M.; supervision, K.G.L., M.K.T., H.S.E.C., A.F. and K.T.K.T. All authors have read and agreed to the published version of the manuscript.

**Funding:** This research was funded by Universiti Malaysia Sabah (UMS) under UMS Research Grant Scheme, grant no. SLB2112 and DN20086.

**Institutional Review Board Statement:** Not applicable.

**Informed Consent Statement:** Not applicable.

**Data Availability Statement:** No data were used to support this study.

**Acknowledgments:** This work was supported by Universiti Malaysia Sabah (UMS) under UMS Research Grant Scheme, grant No. SLB2112 and DN20086.

**Conflicts of Interest:** The authors declare that there is no conflict of interest regarding the publication of this paper.

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
