# Peer review of "Research on Risk Detection of Autonomous Vehicle Based on Rapidly-Exploring Random Tree"

_computation, doi:10.3390/computation11030061_

Round 1

Reviewer 1 Report

This manuscript discusses the development and implementation of the AV-RRT algorithm to address the low practicality and low-efficiency problems of existing path-planning algorithms used in autonomous vehicles. The idea is very good.

I have some minor observations:

- References need a doi number because it is very difficult to find some of them to check their relevance. For example, I can't find reference 5.

- Figure 2: What does "a" mean in the second diamond?

- Figure 4: State why you use the number 4 for the line detector (fours on the diagonal).

- Line 247: "D is the unit vector along the ?rand direction..."  D is a point, not a unit vector.

"the length of (BC)  is t times (BD)..."

It is not mathematically correct. When a vector is multiplied by a scalar t, then the magnitude of the vector (BC)  changes according to the scalar t, but the direction of the vector remains unchanged. In your case, the vectors (BC) and  (BD)  have different directions.

- Is it possible to modify figure 6 a bit to make equation 5 clearer? Also, my previous observation should be added.

Line 452: "Figure 11 shows the map after swelling. " In my opinion, it is not Figure 11 but Figure 9.

Author Response

请参阅附件。

Reviewer 2 Report

Dear Authors, manuscript entitled "Research on Risk Detection of Autonomous Vehicle Based on Rapidly-Exploring Random Tree"
is written very well, this refers to the structure of the paper, the relevance of the content, the contribution to the field as well as the technical processing. 

I suggest only minor changes/corrections and these are: 

- at the end of the Introduction section a short paragraph announcing the content to follow in the following sections of the paper would be useful. For example -  The rest of the paper is organized as follows: Section 2 brings ..., This is followed by ... Finally, the conclusion  ... // this is just a suggestion. 

- equation (12) and (19), coordinates of the center of turning: is this zero (0) or big (O) in the index/subscript, please check. 

- align references according to journal formatting guidelines e.g.: 
   ref_1: spaces are missing between author names ... year, issue,pp; 
   ref_2: similar to previous plus the spacing beetween lines (too big); 
   ref_3: why all capital letters for ZHANG?;
   ...
   ref_13: paper pages are written as 107784-  (what is the last p. number?);  
   ref_14: put conference title in italic (like it is in the rest of the refs); 
   ref_17 and other: the year is not bold?;
   ref_20: what is number at the end representing??
   >> check all the refs and align the style. 

Reviewer 3 Report

General:

-          The Article is well organized and structured, and focuses on autonomous driving vehicles trajectory management and obstacle collision risk avoidance

-          Some issues regarding English and understanding of some phrases have been noticed during read. The paper needs a review of English and some minor corrections.

-          The authors should also give some information regarding the processing power required by the newly proposed algorithm, compared with similar algorithms. Does it need more powerful, or faster processors? Critical information processing is crucial in autonomous vehicles driving, and unless there is a connected vehicles environment, where edge computing might be employed to reduce tasks in the main driving processor, in complicated and rapidly changing traffic situations, this could lead to overloading the processor. (- referring to lines 133-136)

-          Figure 3 needs more explanations in the body text. Does the color of the lines represent something?

-          Does the new algorithm for detection of lane lines based on epirelief curve need that the carriageway is always painted in white, or is it able to detect edges of the carriageway without this type of marking? (lines 191-194)

-          Figures 21 to 24 are not commented in the main text.

Why are there some expressions where you refer to yourselves as “the author”?

Recommendations:

- Please try to add a small comparison between the advantages and disadvantages of the proposed solution, compared to similar ones.

- The Conclusion section may also contain recommendations for industry sectors that may make use of your solution.

Reviewer 4 Report

Dear Editor,
I have read the manuscript titled "Research on Risk Detection of Autonomous Vehicle Based on Rapidly-Exploring Random Tree" submitted for consideration to Computation journal. The paper proposes an improvement of Rapid Exploration Random Tree method for path planning in obstacles present environment. The presentation of the proposed method is clear and precise, and, overall, the paper looks decent. However, I found issues in the presentation of the results that require some paper modifications. Hence, I recommend a minor revision.

1) The reviewed literature is not complete. Please consider some state-of-the-art approaches using potential field, ergodic, and model predictive control (receding horizon) for motion control or path planning methods.

2) The visualization of the obtained path for sparse and moderate obstacle cases is not provided.

3) A much stronger confidence in the success of the method would be provided by a comparison with other algorithms or a comparison with already published test examples. (see comment 4 below)

4) Review and consider related autonomous vehicle control published in https://doi.org/10.1016/j.engappai.2022.105441
Could you utilize your method to provide multi-obstacle test examples?

5) Could you provide the idea (for future research) of how the trajectories could be smoothed or/and shortened? Can the proposed method be extended with a motion constraint, such as maximal curvature / minimal turning radius?

6) The figures show the "Number of experiments" on the x-axis. As far as I can tell this is a Monte Carlo simulation, where multiple simulations are conducted with varying initial and final points. If so, the lines connecting those samples in the visualizations are misleading. There is no dependence between those runs, nor is the order of any importance- I suggest using a bar visualization (two bars per sample, RRT and AV-RRT). Furthermore, I believe some (if not all) presented results could be normalized (scaled according to the distance from the initial to the final point). Then the mean or median could be justifiably calculated.

7) The captions of the figures are extremely short. A reader expects a relatively complete description of the figure he is looking at. Not to browse the main text in order to find explanations for the figures. Please extend the captions to a couple of sentences.
